# God(s)' Mind(s) across Culture and Context

Rita Anne McNamara 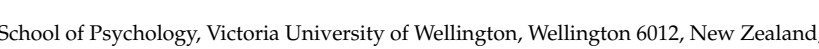

School of Psychology, Victoria University of Wellington, Wellington 6012, New Zealand;
rita.mcnamara@vuw.ac.nz

**Abstract:** This paper explores dimensions of culture and practice that shape the cognitive pathways leading to different beliefs about God(s)' mind(s). Varying socio-ecological sources of insecurity are linked to types and modes of cognitive processing, which in turn promote different constellations of beliefs about supernatural agents dubbed the heuristic and non-heuristic models of God(s)' mind(s). The heuristic model is suggested to take prominence when relatively few cognitive resources are available to devote to thinking about God(s)' mind(s); these conceptions of God(s) should be shaped by the socio-ecological pressures believers face. Conversely, when cognitive resources are available, differences in modes of processing (experiential-intuitive vs. analytical-rational) lead to different mystical and theological/philosophical models of God's mind as a product of more deliberate, effortful processing. By linking beliefs to socio-ecological influences, this paper suggests phenomenological experiences of the supernatural vary across societies as a direct function of the diverse environmental constraints in which people. By linking belief to socio-ecological pressures individuals in societies face, this approach provides a bridge between the intrinsic meaning systems within communities of belief and the cognitive evolutionary approach to parsing the diversity of belief across societies.

**Keywords:** God's minds; culture and cognition; dual processes; theological correctness

## 1. Introduction

Cognitive and evolutionary science has long been in tension with culture, often adopting the assumption that cultural variation is not inherently meaningful but simply a deviation around a universal (conveniently Western) cognitive core substrate (Henrich et al. 2010; Rad et al. 2018; Sinha 2002; Kline et al. 2018). This background assumption of the baseline universality of experiences rooted in the cultures and environments of the West from which most researchers and research methodologies in the existing literature on the cognitive and evolutionary roots of religion originate leads to a pervasive blindness to the phenomenological meanings and lived experiences of religious believers and practitioners. Of particular note for the purposes of this work, cognitive science of religion (CSR) and some evolutionary approaches to religion theorize that religious belief emerged in association with other cognitive mechanisms that evolved for understanding human minds (Boyer 2001; Barrett 2004; Guthrie 1995; Norenzayan et al. 2016; Szocik and Van Eyghen 2021). Human minds, however, do not operate in a vacuum; they interact with our socio-ecological cultural worlds, forming a foundation from which our perception of our own minds, other humans, and nonhuman (including supernatural) beings grows. As such, religious beliefs and practices can be seen as manifestations of embodied cognition that is embedded within the intersection between human and ecology via cultural adaption and transmission mechanisms.

Existing CSR and evolutionary religion research mentioned above also often distinguishes between the God of theologians (conscious God concept) and laity (unconscious God image). Work on "theological incorrectness" suggests that when believers make errors in reporting what God can know and do that violate what theology teaches them to believe, then these mistakes should be in the direction of making God more like a human (Barrett and Keil 1996; Slone 2004). However, the ways human and supernatural minds

are perceived also varies across cultures (Willard and McNamara 2019; Gervais 2013; Gray et al. 2011).

Research on the adaptive functions of religion highlights religion's ability to help groups resolve socio-ecological challenges (Lansing and Fox 2011; Norenzayan et al. 2016; Purzycki and Sosis 2011). Further, supernatural beliefs, religious beliefs, and religious practice have long been thought to act as palliatives in situations of anxiety, uncertainty, and existential terror (Krause 2005; Malinowski 1948; Vail et al. 2010; Kay et al. 2009). Therefore, bringing sociological and ecological pressures into the equation can give greater clarity to how and why believers' differing phenomenological experiences of the supernatural arise in different conditions.

This paper aims to fill the gap between belief as ecological adaptation and belief as a product of cognitive function by reviewing evidence for how particular cognitive processes may be triggered in given socio-ecological pressures. This then gives a broader picture as to how beliefs and believers' phenomenological experiences of the supernatural may vary across cultural boundaries. These ecological and cognitive dynamics then provide the foundation upon which cultural evolutionary processes can operate to produce diverse traditions that either persist, spread, or diminish across time and space.

## 2. Belief as a Product of thought: Heuristic and Non-Heuristic Models of God's Mind

According to the predictive processing theory of perception, we do not see the world as it is but as our brains believe it to be; these beliefs—or models—are based upon accumulated observations that we use to inferentially build persistent ideas about what the world is like (Seth 2014; Clark 2013; Andersen 2017). These inferential models—schema and scripts—serve as mental templates for cognitive heuristics as short-cuts to speed up decision making (Bicchieri and McNally 2018; Leung and Morris 2015; Tversky and Kahneman 1974).

These foundational models of the world are then called upon to interpret each experience. Dual-process models of cognition suggest the ways we think through these models of the world fall within a few broad categories: fast/easy (a.k.a. 'type 1') vs. slow/effortful (a.k.a. 'type 2') and within type-2 processing, experiential–intuitive vs. rational–analytic (Epstein 1998; Shiloh et al. 2002; Evans 2008; Neys 2021). Type 1 processing operates below the level of alert awareness and does not require many cognitive resources. Type 2 processing, on the other hand, requires more effortful control and often happens within the level of awareness. As such, type 2 processing capacity is often rapidly depleted when organisms must divert most or all their attentional effort toward navigating highly threatening, existentially insecure environments (Mullainathan and Shafir 2013). Conceptions of God's mind (i.e., what believers expect God(s) to think, attend to, act on, and care about; the content of God(s) minds) may be more impacted by these below-the-level-of-awareness (i.e., heuristic) processes when most thinking about the supernatural is restricted to type 1 processing, as happens in environments with relatively higher degrees of existential threat (McNamara and Purzycki 2020). For example, a believer may think, 'God is too busy looking after the universe to care about my problem.' This belief reflects a bias toward seeing God as having human-like limitations, which arise as a product of this believer applying social cognition developed for understanding human agents without additional type-2 processing for reflection to expand their thinking to conceptualize the mind of an infinite being. However, focusing strictly on high insecurity only covers part of the spectrum of human experience. This paper also discusses non-heuristic models of God's minds: mystical and theological/philosophical, which are shaped by different, parallel-competitive modes of type 2 processing: experiential–intuitive vs. analytical–rational (Epstein 1998; Hodgkinson et al. 2009; Cacioppo et al. 1996; Taves 2020).

## 3. Heuristic Models of God's Mind: Control, Parochialism, Social Conservatism

The following sketch of God's mind under conditions of high socio-ecological/psychological insecurity and uncertainty suggests that within the heuristic model of God's mind, God is believed to be in control of conditions contributing to human life and well-

being; parochial (interested in believers' in-group); interested in correct ritual adherence; supportive of existing social structures; and anthropomorphized to suit application of social cognitive mechanisms that initially evolved to understand human agents. Importantly, these beliefs may arise regardless of the formal doctrinal theology of the particular religious system. This is because these heuristically driven, theologically incorrect beliefs are here theorized to be cognitively more readily available without the additional processing and reflection necessary to arrive at a theologically correct belief (Slone 2004; Barrett and Keil 1996).

### 3.1. God Is in Control: Psychological Insecurity from Violated Expectations, Type 1 Processing, and Managing Cognitive Resources

Situations that challenge the integrity of meaningful experiences of the world (Heine et al. 2006); reminders that their bodies and lives are finite (Burke et al. 2010; Shepherd et al. 2011); and reminders of limited personal control (Kay et al. 2010a) all suggest that beliefs about God's control help mitigate psychological insecurity (Kay et al. 2010b; Shepherd et al. 2011). Therefore, when experiencing psychological insecurity due to violated expectations, believers should be particularly prone to reporting belief in a controlling God (McNamara and Purzycki 2020).

### 3.2. God Has Chosen Us: Socio-Ecological Insecurity from Beyond the In-Group

One key source of existential threat that depletes type 2 processing capacity comes from objective threats to survival. Neurocognitive mechanisms have evolved for assessing and reacting to threat (Hinds et al. 2010; Nesse 2005). Though threat detection systems should evolve to minimize error, they should be biased towards mistaking non-threats for threats (false positives) over missing a true threat (false negatives) to minimize the cost of unavoidable error in an imperfect system. They may preferentially bias an organism towards interpreting neutral or ambiguous stimuli as threatening (Bateson et al. 2011; Flannelly et al. 2007; Stein and Nesse 2011) and can be activated by actual, present threats as well as inferred, potential threats (Boyer and Lienard 2007).

Social and physical threats form distinct environmental–ecological issues that must be navigated for survival (Stein and Nesse 2011; Woody and Szechtman 2011). The social dimension presents different kinds of responses to traumatizing events (e.g., combat experience more often results in post-traumatic stress disorder than wildfires, see: (Bracha 2006). Social adaptive challenges can be further divided into threats from pathogens, from other humans outside one's social group, and from others within one's social group (Fincher and Thornhill 2012; Neuberg et al. 2011). These systems have been used to explain broad patterns in religious practice and belief (Johnson et al. 2014).

The social repercussions of non-social environmental existential insecurity has largely been linked to societal-level differences in tolerance for norm violations and the strength of informal social controls (Fincher and Thornhill 2012; Gelfand et al. 2006; Hruschka et al. 2014; Van de Vliert 2008). Greater exposure to threat pushes cultures towards adopting 'tighter' cultural forms with more focus on following norms and less tolerance of social deviance, including around religious practice and belief. This trend towards increasing norm adherence and religiosity as insecurity increases has been documented extensively, especially in the absence of formal, non-religious institutions (Norris and Inglehart 2004). Pathogen exposure can be both a function of the non-social environment and easily exacerbated by contact with other humans (Fincher and Thornhill 2012). The behavioral immune system is thought to underlie psychological responses to cues of 'diseasiness' (Curtis et al. 2011), which include xenophobia, conformity, lower openness to experience, and lower extraversion (Fincher and Thornhill 2012; Faulkner et al. 2004). Disgust is the emotional cue that triggers conscious response when the behavioral immune system is engaged (Curtis et al. 2011). Ritual behavior itself may have evolved as a means of dealing with issues of contagion threat (Boyer and Lienard 2007). Disgust has further been co-opted into social, moral, and political decision-making (Inbar et al. 2012; Jones and Fitness 2008; Russell and

Giner-Sorolla 2013), which sustain religious norms of purity and intolerance for deviance away from accepted levels of purity. Those who endorse conservative political values have been shown to value purity more, to show greater disgust sensitivity, and to have a higher moral disgust response (Graham et al. 2009; Helzer and Pizarro 2011; Inbar et al. 2012). Social threats associated with potential pathogen exposure remain distinct from social threats from the behavior of other humans. Inter-group hostilities through warfare are thought to be as old as humanity (Choi and Bowles 2007; Lahr and Foley 1998; Turchin 2011). Personal experience with heavy intergroup violence is linked to greater emphasis on local and kin groups, more risk aversion, and greater willingness to incur costs to punish antisocial in-group members (Bauer et al. 2014; Callen et al. 2014).

Taken together, external threats from non-social environmental instability, pathogen stress, and intergroup violence all suggest that people's responses tend towards greater focus on local in-groups. What does this imply for the mind of God? Supernatural agent beliefs in societies facing these kinds of socio-ecological pressures should include higher endorsements of belief that supernatural agents favor the faithful above and beyond the rest of humanity (a parochial God), thus supporting in-group favoritism and potentially out-group derogation. Further, supernatural agent beliefs should include that they care deeply about proper ritual performance and public displays of faith (Boyer and Lienard 2007; Purzycki and Arakchaa 2013). In cases where external socio-ecological threats also boost conformity, supernatural agents should also be believed to care about believers' submission to existing social norms of propriety. This emphasis on propriety might be construed as purity (Haidt 2012). In places where a heavy disease burden has made a noticeable impact on supernatural beliefs, we should see an increase in belief that supernatural agents care about maintaining food and hygiene taboos. This moralized dimension of contagion purity is present in living religious traditions, especially in the Indian subcontinent (Armstrong 2006; Rozin et al. 1999; Shweder et al. 1997). Therefore, societies with heavy disease burden and reminders of contagion should promote greater belief that supernatural agents care about maintaining hygiene, food purity taboos, and cleanliness rituals.

### 3.3. God Is a Reflection of Society: Socio-Ecological Pressures from the In-Group, Cultural Strategies, and Societal Structure

Social theories of religious beliefs and practices propose that religion is a product of the way society is structured, how individuals interact within society, and the aspects of society that either facilitate or challenge prosocial interaction (Durkheim 1995; Geertz 1957; Purzycki 2010; Sosis and Ruffle 2004; Norenzayan et al. 2016). Social norms, including religious beliefs and practices, are internalized by individuals within societies as they navigate their social environment (Sripada and Stich 2006; Leung and Morris 2015; Bicchieri and McNally 2018; McNamara and Purzycki 2020). Socio-ecological stresses from in-groups are shaped by the society's dominant cultural strategy, or the constellations of norms and institutions used to structure daily life (McNamara and Purzycki 2020). Strategies that develop when social networks are diffused and have poor formal institutions of social control are predicted to promote belief that God cares about maintaining an honorable reputation; punishes theft; does not punish acts of violence enacted against thieves or other persons threatening one's honor; watches believers; and cares about maintenance of sexual and gender norms. Strategies that develop in societies with stable, cooperative social hierarchies are predicted to bolster the belief that God wants believers to be humble; cares that believers fulfill the obligations and duties inherent to their place in the hierarchy; exists as an extension of the earthly hierarchy in human society; cares about correct ritual performance; and punishes violations of social norms that uphold the traditional hierarchy. Strategies that develop in societies with extensive individual autonomy in large, anonymous, complex societies with strong, effective non-religious institutions for social control allow individuals to form separate, bounded selves independent from the social networks they inhabit. This separation of self from group shifts the focus from proper external display of religious commitment to proper internal motivation and belief in direct

contact with God. These societies should therefore exhibit the belief that God cares about individuals, does not care about proper ritual performance; does not care about hierarchies; cares about correct content of internal devotion to God; and is kinder and more benevolent than punitive and harsh (McNamara and Purzycki 2020).

## 4. Non-Heuristic Models of God's Mind

Most research on how religious beliefs arise from psychological processes focuses on the 'naturalness' of religion, generally focusing on how religious belief is effortless and beyond conscious control—i.e., the heuristic model of God's mind. This implies that belief in supernatural agents that contradict the heuristic model will necessarily require effortful, type-2 thinking (e.g., Barrett and Lanman 2008; Bering 2010). However, how might Gods' minds change as more effortful, type 2 thinking capacity is applied to thinking about them?

Mystics and theologians are most likely to think deeply about the nature of the divine and are therefore likely to hold a non-heuristic model of God's mind. Both often describe God as abstract and share a denial that God has anthropomorphic traits, but their descriptions of God's mind differ (Armstrong 1993; Kroll and Barchrach 2005). These mystic vs. philosophical/theological models of God's mind may differ according to different emphasis on the two main modes of type 2 processing: experiential–intuitive and analytical–rational. These modes differ from type 1 vs. type 2 processing in that both modes can contribute to conscious experience at the same time (Evans 2008). Experiential–intuitive and analytical–rational processing modes have been shown to receive different preferred, normative proportions of type 2 processing capacity across individuals and cultures (Buchtel and Norenzayan 2009; Choi et al. 2007; Epstein et al. 1996). Emphasis on either mode of processing can also be seen in the differences between mystical vs. theological/philosophical models of God's mind.

### 4.1. God Is Everything: Intuitive–Experiential Thinking, Altered States of Consciousness, Mystical Experience, Awe, Absorption, Flow, and Mindfulness

God as perceived in mystical experiences is often abstract, amorphous, and intimate (Armstrong 1993; Kroll and Barchrach 2005). A growing body of work on altered states of consciousness suggests intuitive–experiential processing is implicated in these experiences. The hunt for God in the brain has led to evidence for various neural substrates behind mystical experiences (e.g., Beauregard and Paquette 2006; Persinger et al. 2010; Taves 2020; Cristofori et al. 2016; Deane 2020; Lancelotta and Davis 2020). Mystical experience is often associated with a sense of separation from an everyday, ordinary sense of self (Yaden et al. 2017; Krause 2018; Deane 2020; Millière et al. 2018). Heightened intuitive–experiential processing and reduced sense of self can be associated with a sense of extreme threat and uncertainty (Yaden et al. 2017; Whitehouse 1996) as well as safety and certainty (Beauregard and Paquette 2006).

The neurophysiology of altered states of consciousness taps into various aspects of mystical experience—especially as these mystical experiences relate to an altered, porous, expansive self or dissolved self (Taves 2020). The Default Mode Network has come under particular focus in the study of altered states and of social cognition, as this network of neural activation appears to both have general implications for thought left diffused when not focused on particular tasks, when focused on simulating states of mind in others, and the beyond-the-mundane experiences of expansive/dissolving self in altered states of consciousness (Smigielski et al. 2019; Palhano-Fontes et al. 2015; Raichle 2013). Thus, when thinking in a primarily experiential, integrative mode, believers may likely report belief that God is present in close physical proximity while simultaneously feeling less tied to and bounded by their own bodies. Such a model of God's mind might feature a sense that God and self are fused and connected with the wider surroundings.

Mystical experience may also arise from more mundane emotions and mental states of awe (Keltner and Haidt 2003; Shiota et al. 2007; Van Cappellen and Saroglou 2012), absorption (Tellegen and Atkinson 1974; Luhrmann et al. 2010, 2021), flow/peak experience

(Csikszentmihalyi 2014; Moneta 2004), and mindfulness (Grant and Zeidan 2019; Sedlmeier 2018; Howell et al. 2011). Like altered states of consciousness, these emotions and mental states also feature an altered sense of self and focused attention on an immediate, present target. Situations that evoke emotional/mental states, such as awe, absorption, flow, or mindfulness (as are constructed in many traditions of ritual and religious practice), share a common theme of activating neurocognitive processes that in turn amplify intuitive–experiential processing.

This evidence suggests experiential–intuitive type 2 processing will shape the mind of God into an abstract, disembodied presence. A more fluid sense of self may promote the belief that God is intimately present—suffused throughout the believer's sense of self and throughout the universe. The sense of God's immensity or dissolved self could also lead to a belief that God is terrifying (Keltner and Haidt 2003; Proulx et al. 2012). At the same time, a focus on immediate experience might also promote the belief that God is non-temporal and does not care about human time-bound concerns. Thus, a mystical model of God's mind might promote pantheism or belief in God as a non-anthropomorphic force pervading the universe.

### 4.2. God Is the Unmoved Mover: Rational–Analytic Thinking

One theological stance that is not often considered in this framework is non-theism (O'Grady and York 2012). Non-theism itself can be a catch-all term for everything from an asserted belief that God does *not* exist to the belief that God *does* exist but is not human-like. Non-theism has been used to describe both Buddhism and certain groups of Quakers (Glasenapp 1966; Riemermann 2006) and include pantheism and deism (Pennycook et al. 2012).

The theological/philosophical model should be dominated by the rational–analytic mode of type 2 processing. However, analytical thinking is often equated to full denial of God's existence (Baimel et al. 2021; Kalkman 2014). Two ways of conceptualizing a non-heuristic God that have shown up in philosophical/theological religious thought are the Hidden God and the Absent God (Ebeling 1964; Elders 1990). Importantly, both are forms of deism that suppose God was the ultimate cause of the natural world but is not currently active in it. Deistic models of God's mind also differ from pantheistic models of God's mind in that deism may be more closely associated with analytical processing (Norenzayan and Gervais 2013; Pennycook et al. 2012).

### 5. Discussion

Beliefs about God(s)' mind(s) are predicted to differ as a function of variation in socio-ecological conditions, which promote different modes of cognitive processing, which in turn feed into different phenomenological experiences and perceptions of the supernatural. These beliefs constitute different models of God(s)' mind(s): a heuristic model dominated by fast, effortless type 1 processing; a mystic model dominated by effortful experiential–intuitive type 2 processing; and a philosophical/theological model dominated by rational–analytical type 2 processing. Non-heuristic models of God(s)' mind(s) arise mainly in conditions that allow more type 2 processing to be devoted to thinking about God, as is the case for religious professionals, such as mystics and theologians.

One of the main strengths of this approach is its ability to provide a framework for understanding to interpret the socio-ecological roots of meaning systems across lay and specialist practitioners in diverse traditions. This provides a bridge between the existing work on cognitive science of religion and the more lived experience of religious life among practitioners by linking the patterns of belief to broader trends of ecological adaptation. It capitalizes on the power of cultural evolutionary processes to examine how psychological functioning in various environments may promote patterns of belief and make room for more nuanced interpretation of the lived experience of religion through the rigor of a cognitive scientific lens. This also provides a framework to understand when and how

religious specialists would have beliefs that deviate from lay persons and may indicate underlying patterns in variation across beliefs both within and across traditions.

*Implications and Limitations*

Part of the limitation in this work stems from the continued Western bias in the research behind cognitive processing itself (Rad et al. 2018). Individualism and brain focus within the Western model of mind (or the intuitive/unreflective set of beliefs and assumptions about how minds work) in general and in scientific approaches to the mind in particular make the extent to which cognition itself might be modulated by cultural forms difficult to pinpoint given the current state of the literature (Lillard 1998; Luhrmann 2011; McNamara et al. 2021). Placing the self as the foundation for inferring God's mind assumes a sense of self, as defined within the Western experience, is the most basic aspect of psychological experience, but cross-cultural studies of self indicate the Western, individualistic self is not universal (Broesch et al. 2011; Heine 2001; Markus and Kitayama 1991; Lysenko 2017). Similarly, the sense of self as separate has been proposed as the basic tenant of secularism (Taylor 1989, 2007). The secularized self—as opposed to the porous, non-secular self—may be unique to Western cultures (e.g., Leung and Cohen 2011; Luhrmann et al. 2021) and may be the product of a unique Western cultural, religious tradition (Asad 2003; Laine 2014; Masuzawa 2005).

Importantly, the model of self assumed in Western settings may be less readily accessible to people living in conditions of extreme material, existential insecurity. Various lines of research across psychology, anthropology, and neuroscience describe ways that the distinction between self and not-self can become blurry in situations of extreme psychological and physiological duress. These situations that promote a more porous sense of self can be induced unintentionally through actual physical threat (e.g., battlefield stress and spiritually uplifting experiences of elevation and awe: (Yaden et al. 2017)) or physical disturbance of neurobiological functions (brain trauma: (Taylor 2006); ecstatic epileptic seizures: (Hansen and Brodtkorb 2003; Ogata and Miyakawa 1998; Tucker et al. 1987)). The sense of self as starkly separated from the rest of reality can also be reduced by manipulating neurobiological function through practices, such as asceticism, meditation, and psychotropic plant medicines (Armstrong 1993; Flood 2004; Kroll and Barchrach 2005; Taves 2020; Millière et al. 2018). Further, individual differences may predispose some people towards experiencing a more porous self, as with schizotypal spectrum traits and underlying predispositions towards non-normal experiences often associated with either clinical or sub-clinical diagnoses of psychosis (Fabrega 1989; Polimeni and Reiss 2003; Genovese 2005; Willard and Norenzayan 2017).

Rather than giving specific examples from the ethnographic literature, this paper provides predictions of broad patterns in belief and supernatural phenomenological experience that one might expect in given cognitive and socio-ecological conditions. More work from indigenous perspectives from within these traditions and in collaboration with cognitive sciences researchers needs to be done to further assess how many of these patterns are demonstrated in the real world. Further, the dynamics of cultural evolution do not necessarily mean that a given socio-ecological stressor will produce a given cognitive effect immediately. Rather, humans tend to carry their beliefs with them, which—according to predictive processing—can fundamentally shape their experiences and reactions to the new environments in which they find themselves. Therefore, additional work to examine patterns of religious and spiritual belief change as societies move and intermix through migration, colonization, missionization, and other intercultural contact events can also show the underlying dynamics of how culture directly shapes cognition and vice versa (McNamara et al. 2021).

## 6. Conclusions

By incorporating a perspective on how external influences shift psychological experience, we can gain a more precise understanding of the motivations behind both practice

and belief. In doing so, we can make sense of seemingly contradictory phenomena as part of the broader story of human nature. This approach also has the potential to more accurately capture the true scope of variation across belief systems, as individual beliefs may arise as a result of the combination of cognitive processing within particular environments. Importantly, this can aid in reducing the existing bias toward centering Western ways of thinking by focusing on the within-context adaptive connections between beliefs as they operate in the broader socio-ecological and cognitive systems across societies.

**Funding:** This research received no external funding.

**Institutional Review Board Statement:** Not applicable.

**Informed Consent Statement:** Not applicable.

**Data Availability Statement:** Not applicable.

**Acknowledgments:** I thank Benjamin Purzycki for his helpful comments on previous versions of this paper.

**Conflicts of Interest:** The authors declare no conflict of interest.

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
