# Peer review of "God(s)’ Mind(s) across Culture and Context"

_religions, doi:10.3390/rel14020222_

Round 1

Reviewer 1 Report

Thank you very much for an enjoyable article. Please find my comments in the attached file.

Author Response

I thank this reviewer for their thorough and helpful comments. My responses by section are listed below:

Section 1:

The reviewer makes a fair point about the human distinctiveness of experiences that are not captured in this paper. My focus here is on what the interpretation of perceptual experiential states has on beliefs as they are applied within the meaning systems humans operate within. A further exploration of the perceptual level of these experiences is beyond the scope and space of this piece.

Section 2:

It is a common mis-conception that type-2 processing is strictly rational-logical. By the definition I use here, any thought process that entails effortful and intentional focus falls under the type-2 umbrella. Take, for example, meditative processes that practitioners effortfully and intentionally engage with to enduce states of consciousness that allow for non-linear and non-rational inspiration. These have the same attentional demands as rational-analytical thinking, but engage different neural systems.

I added an example of a belief that may arise that demonstrates how God's mental capacities may be perceived as being more human-like than would be logically true of an infinite being if a believer were only using social cognition as developed for understanding human agents without additional type-2 processing to expand their thinking.

I added the line: (i.e. what believers expect God(s) to think, attend to, act on, and care about; the content of God(s) minds) as a more direct definition of what I mean by God's minds. I hope this clarifies the confusion the reviewer identified. 

Section 3:

I see the point in allowing for the nuance that this reviewer is making. However, the heuristic model should arise whenever the conditions make this pattern of cognitive processing most accessible, regardless of the faith tradition. I added the following:

Importantly, these beliefs may arise regardless of the formal doctrinal theology of the particular religious system. This is because these heuristically-driven, theologically incorrect beliefs are here theorized to be cognitively more readily available without the additional processing and reflection necessary to arrive at a theologically correct belief (Slone 2004; Barrett and Keil 1996).

to make the point about theological incorrectness of these beliefs more clear.

The confusion about whether God cares about purity rituals is hopefully clarified with the additional definition added to Section 2. The belief that God cares about members of these communities doing the rituals is what is being predicted here. This belief that god cares about these rituals would add further behavioural motivation to enact them above and beyond the group cohesive aspects of these actions the reviewer has already identified.

Section 4:

I hope that the additional definition in Section 2 clears this up.

Section 5: 

Non-Western conceptions of the self are indeed important across the societies they are indigenous to. The matter being argued here is that the cognitive and psychological study of religion as a whole is biased toward assuming the Western model is the baseline. Further basic cognitive and psychological science needs to be done to establish what other downstream consequences these variations in sense of self may have for systems of belief within religious traditions and across all areas of life where social condition and self-concept are involved. 

Thanks again to this reviewer for their encouraging words, I too hope to see more of this research in the future.

Reviewer 2 Report

Well researched and well written article. It is very relevant and up to date. The focus on cognitive and evolutionary sciences fits in well with the focus of the article. The integration and referencing to classic work and more accessible works are especially valuable in this article. As the author states, the approach allows for a 'understanding to interpret the socio-ecological roots of meaning systems across lay and specialist practitioners in diverse traditions'. 

I found this article highly interesting and it was a pleasure to review it. My only recommendation is that the author will add a sentence or two at the conclusion.

Author Response

Thanks to this reviewer for their supportive and encouraging comments on this draft. The following has been added to the conclusion:

This approach also has the potential to more accurately capture the true scope of variation across belief systems, as individual beliefs may arise as a result of the combination of cognitive processing within particular environments. Importantly, this can aid in reducing the existing bias toward centering Western ways of thinking by focusing on the within-context adaptive connections between beliefs as they operate in the broader socio-ecological and cognitive systems across societies.